# Cross border population movements across three East African states: Implications for disease surveillance and response

**Patrick King** [1]*, **Mercy Wendy Wanyana**[1], **Harriet Mayinja**[2], **Brenda Nakafeero Simbwa**[1], **Marie Gorreti Zalwango**[1], **Joyce Owens Kobusinge**[2], **Richard Migisha**[1], **Daniel Kadobera**[1], **Benon Kwesiga**[1], **Lilian Bulage**[1], **Doreen Gonahasa**[1], **Peter Ahabwe Babigumira**[3], **Serah Nchoko**[4], **Edna Salat**[4], **Freshia Waithaka**[4], **Oscar Gunya**[4], **Fredrick Odhiambo**[4], **Vincent Mutabazi**[5], **Metuschelah Habimana**[5], **Gabriel Twagirimana**[5], **Ezechiel Ndabarinze**[5], **Alexis Manishimwe**[5], **Harriet Itiakorit**[3], **Samuel Kadivani**[4], **Katy Seib**[6], **Ellen Whitney**[6], **Alex Riolexus Ario**[1]

1 Uganda Public Health Fellowship Program, Uganda National Institute of Public Health, Kampala, Uganda, 2 Division of Surveillance, Information and Knowledge Management, Ministry of Health, Kampala, Uganda, 3 Department of Global Health Security, Baylor College of Medicine Children's Foundation, Kampala, Uganda, 4 Field Epidemiology and Laboratory Training Program, Kenya National Public Health Institute, Nairobi, Kenya, 5 Field Epidemiology Training Program, Ministry of Health, Kigali, Rwanda, 6 International Association of National Public Health Institutes, Emory University, Atlanta, Georgia, United States of America

* kingp@uniph.go.ug

**Data Availability Statement:** The datasets upon which our findings are based belong to the Uganda

## Abstract

The frequent population movement across the five East African Countries poses risk of disease spread in the region. A clear understanding of population movement patterns is critical for informing cross-border disease control interventions. We assessed population mobility patterns across the borders of the East African states of Kenya, Uganda, and Rwanda. In November 2022, we conducted Focus Group Discussions (FGDs), Key Informant Interviews (KIIs), and participatory mapping. Participants were selected using purposive sampling and a topic guide used during interviews. Key informants included border districts (Uganda and Rwanda) and county health officials (Kenya). FGD participants were identified from border communities and travellers and these included truck drivers, commercial motorcyclists, and businesspersons. During KIIs and FGDs, we conducted participatory mapping using Population Connectivity Across Borders toolkits. Data were analysed using thematic analysis approach using Atlas ti 7 software. Different age groups travelled across borders for various reasons. Younger age groups travelled across the border for education, trade, social reasons, employment opportunities, agriculture and mining. While older age groups mainly travelled for healthcare and social reasons. Other common reasons for crossing the borders included religious and cultural matters. Respondents reported seasonal variations in the volume of travellers. Respondents reported using both official (4 Kenya-Uganda, 5 Rwanda-Uganda borders) and unofficial Points of Entry (PoEs) (14 Kenya-Uganda, 20 Uganda-Rwanda) for exit and entry movements on borders. Unofficial PoEs were preferred because they had fewer restrictions like the absence of health screening, and immigration and customs checks. Key destination points (points of interest) included: markets, health facilities, places of worship, education institutions, recreational facilities and business

Ministry of Health and Uganda National Institute of Public Health (UNIPH). They are publicly available upon request from the Uganda National Institute of Public Health through info@uniph.go.ug.

**Funding:** The project was supported by the International Association of National Public Health Institutes (IANPHI), Uganda National Institute of Public Health, and the United States Centres for Disease Control and prevention grant number NU14GH001238. The contents of this manuscript are solely the responsibility of the authors and do not necessarily represent the official views of the US-Centers for Disease Control and Prevention, the department of health and human services, Makerere University school of Public Health or the Ministry of Health. The funders had no role in study design, data collection and analysis, decision to publish, or preparation of the manuscript.

**Competing interests:** The authors have declared that no competing interests exist.

towns. Twenty-eight health facilities (10- Lwakhakha, Uganda, 10- Lwakhakha, Kenya, and 8- Cyanika, Uganda) along the borders were the most commonly visited by the travellers and border communities. Complex population movement and connectivity patterns were identified along the borders. These were used to guide cross-border disease surveillance and other border health strategies in the three countries. Findings were used to revise district response and preparedness plans by strengthening community-based surveillance in border communities.

## Background

The East African region is threatened by numerous emerging and re-emerging diseases of international concern. These include wild polio, yellow fever, Ebola Virus Disease (EVD), Marburg virus disease, Crimean-Congo haemorrhagic fever, hepatitis E virus, and cholera [1]. Increased cross-border movements of humans and animals could fuel the spread of these diseases with the resultant effect of affecting population health and straining the health systems in the region [2].

Collection of data on population mobility patterns such as volume and destination at Points of Entry (PoEs) is gaining momentum in the East African region. However, this is still insufficient for providing evidence for decision-making. The Population Connectivity Across Borders (PopCAB) methodology provides detailed information on mobility patterns including who, where, when, why, and how of human mobility and community connectivity [3]. Furthermore, the methodology eases the integration of population mobility in public health surveillance, programming, preparedness, and response initiatives.

A clear understanding of the unique population movement patterns is essential for tailoring communicable disease preparedness and response strategies that aim to limit the international spread of disease. Characterisation of movement patterns including destinations, routes used, and reasons for travel could facilitate more accurate quantification of health risks, importing, and exporting of disease [3, 4]. By considering the complex ways in which people move and interact with their environment, public health officials can design more effective preparedness and response strategies.

The Uganda National Institute of Public Health (UNIPH) under Uganda Ministry of Health together with the respective ministries in Kenya and Rwanda conducted a PopCAB activity on the Uganda-Kenya Lwakhakha border and the Uganda-Rwanda Cyanika border. This was done to explore population movement patterns, identify points of interest and travel routes, visualise population movement patterns, and suggest suitable public health recommendations for surveillance and preparedness. The findings would help strengthen tailored interventions to prevent, detect, and respond to the spread of communicable diseases including the EVD outbreak at the time of the assessment.

## Methods

### Study setting

We conducted the assessment at Lwakhakha (Uganda-Kenya border) and Cyanika (Uganda-Rwanda border). The Lwakhakha border is located at Namisindwa District in Uganda and Bungoma County in Kenya. The Cyanika border is located at Kisoro District in Uganda and Burera District in Rwanda. The borderline between Uganda and Rwanda extends from the

tripoint with the Democratic Republic of the Congo in the west to the tripoint with Tanzania in the East spanning a distance of 188 km. Uganda-Kenya borderline extends for 870 km from the tripoint with South Sudan in the north to the tripoint with Tanzania in the south.

We used the PopCAB methodology toolkit developed by US Centers for Disease Control and Prevention to gather and analyse population mobility including characteristics of travellers, reasons for travel routes taken by travellers, travel routes, and key destinations/points of interest [5].

## Participants and sample selection

Twelve Key Informant Interview (KII) participants, eight in Uganda and four from Kenya were purposively selected. Leaders from multiple sectors at the district (in Uganda), and county (in Kenya) level included District Health Officers, District Surveillance Focal Persons, Sub-County Internal Security Officers, Port Health Focal Persons, County Disease Surveillance Coordinators, Immigration Officers, Port Health Officers, and County Community Services Focal Persons.

We conducted Nine Focus Group Discussions (FGD), seven from Uganda and two from Kenya. Participants in the FGD included: community groups such as fisherfolk, truck drivers, boda-boda/cyclist riders, travellers, and businesspersons. Each FGD had 8 persons. All participants in Fisherfolk, truck drivers, boda-boda riders FGDs were male. In the study setting these occupational group are male dominated. FGDs for business persons and travellers were disaggregated by gender.

## Data collection methods and tools

From November 1st–15th, 2022 we conducted KIIs and FGDs using an interview guide. The interview guide collected data on the characteristics of travellers, reasons for crossing the borders, when they cross and means used for travel/crossing the border, influence of public health threats like EVD on movement, recent movement to EVD affected areas. The KII findings generated information utilized in the selection of categories of people to be considered for the FGDs at the borders. Using an FGD guide, we conducted FGDs with border communities and travellers.

All KIIs and FGDs had a participatory mapping component using maps of Uganda-Kenya border and Uganda-Rwanda border. Areas of interest and routes were annotated on the maps by the interviewer with guidance from the participants.

## Data management and analysis

Discussions were transcribed by a note taker and analysis was later conducted using a thematic analysis approach. We developed codes and grouped codes under sub-themes and themes. Themes annotated maps from the various interviews were summarised into a map for each border point (Cyanika and Lwakhakha borders) to provide a comprehensive picture of the routes and PoEs used. We used QGIS software to draw maps.

## Ethical considerations

The study was done in reference to a memorandum of understanding and umbrella protocol (Grant number: NU14GH001238) between the field Epidemiology training programs of the three countries (Uganda, Kenya, Rwanda), this study was waived the process of full institutional review board. In addition, this activity was reviewed by United States Centers for Disease Control and Prevention (U.S CDC) human subjects review board and conducted

consistent with applicable federal law and CDC policy. § §See e.g., 45 C.F.R. part 46, 21 C.F.R. part 56; 42 U.S.C. § 241(d); 5 U.S.C. § 552a; 44 U.S.C. § 3501 et seq.

We obtained written informed consent from the local area leaders and community representatives prior to conducting the study. Verbal informed consent was sought from all participants who were selected to participate in the study and was documented as a yes/no for every record. They were informed that their participation was voluntary and their refusal would not attract any negative consequences. Data collected did not contain individual personal identifiers as a way of ensuring confidentiality.

## Results

### Characteristics of travellers

Respondents reported that mainly individuals below 35 years frequently cross the border. There were differences in the age groups travelling depending on the reason for travel. Both genders travelled across the border. Unique to the Lwakhakha border, respondents in Uganda reported both genders crossing the border while Kenyan respondents reported mostly males crossing the border. Nationalities in the East African region (Ugandan, Kenyan, South Sudanese, Congolese, and Rwandese) commonly travelled across the borders.

*"......Youth and young adults are the most common people moving across both countries......usually men aged 15–30 years and women aged 14–20 years travel for employment opportunities"* **FGD P4, boda-boda cyclist Uganda**

*"…. It depends on the activity, children from 6 to 15 years from Uganda move to Kenya selling snacks and fruits/vegetables….. from 15 to 30 years Ugandans move from Mbarara, Mbale and Namisindwa for casual labour as maids…. adults above 20 to 35 years move for casual labour on farms in Eldoret, Chwele while others move to Nairobi, Eldoret, Chwele to work as maids"* **FGD, P7 boda-boda Kenya**

*"…. Sudan refugees cross weekly and monthly…"* **FGD P1, Uganda**

*"…..Ugandans, Rwandese and Congolese cross Cyanika border as a connecting route to Gisenyi and Goma "……*  **FGD P11, Uganda**

### Reasons for travel

**Livelihood.** Respondents reported trade in various items including food, livestock and household items across the border. They cited cheaper goods on the other side of the border as a motivation to travel to various markets across the border. In all three countries respondents reported travelling in search of employment opportunities, mainly casual work on the other side of the border. Unique to the Cyanika border, communities travelled from Rwanda for mining activities in Uganda, Commercial sex and smuggling were also reported as reasons for travel.

*"…People travel to Kisoro Market on the side of Uganda and Musanze Market on the side of Rwanda to buy and sell different things…"* **Participant 5 FGD Cyanika**

*"…..People cross the border for business. They come and buy farm produce like bananas in markets in Bududa District (Uganda). Others go to Kampala and Jinja to buy items like clothes, shoes etc… some smuggle goods across the border…."* **Participant 5 FGD Lwakhakha**

*".… travelers move to Mubende (Uganda) from Rwanda for gold mining .…Rwandese women who work in bars and also do sex work, others engage in escort services (prostitution at the border, either Cyanika PoE or Bunagana PoE.….."* **P2, FGD Cyanika**

**Religion and culture.** Respondents reported travel to attend various religious and cultural events including church services, pilgrimages, and cultural events such as circumcision.

*".….Ugandans, Kenyans, and Congolese also visit Kibeho (Rwanda) for religious services in August yearly.…there are also a number of Rwandese who travel for the annual Martyrs Day celebrations in Namugongo (Uganda).…"* **Participant 2 FGD Cyanika**

*".….Ugandans also Move to Kenya for festivals of circumcision; they come from Sironko Manafwa, Butiru, Bududa all the way dancing into Kenya and go back to Uganda.…."* **KII Lwakhakha**

**Healthcare.** According to respondents, individuals travel to seek healthcare services on the other side of the border. Reasons for this included more affordable care or even free and specialist services. Communities from Rwanda and Kenya visited Ugandan health facilities near the border for free health services. Respondents in Kenya reported seeking specialist services such as Ophthalmology services in Tororo District (Uganda). Respondents in Uganda reported travelling to Kenya for better maternal health, immunisation, and geriatric services for the elderly in Kenya.

*".…The cost of health services is cheaper in Uganda. In Rwanda people complain that you need to pay for insurance to access medical services and without it they cross over to Uganda for free (Clare Nsenga Health Facility) / cheaper health services (other health facilities).… Rwandese come to Uganda for seeking health care services like antenatal care, delivery, post exposure prophylaxis because those services are free .…"* **Participant 8 FGD Cyanika**

**"**.….the Ugandan women come to Kenya for maternal services and antenatal services.… Kenya offers better packages for delivering mothers.… They always bring under five children because the health services in Kenya are free and they always give mosquito nets to mothers. The mothers who usually cross for health services to Kenya are aged 30 to 40 years…**"** **Participant 8 FGD Lwakhakha**

**Traditional healing services.** Participants also mentioned that travellers visited Uganda for to visit traditional healers as quoted;

*"…there are known traditional healers in Kabindi community and in Nyarusiza sub-county, Bushenyi District in Uganda who are visited by many people from Rwanda…"* ***KII Cyanika***

*"…herbalists/native doctors around Elgon reserve forest in Uganda are visited by people from across the border for traditional healing…"* **Participant 2, FGD Lwakhakha**

**Education.** Education was one of the main reasons for travel. Respondents reported travelling for better and affordable education on the other side of the border of Uganda, Kenya, and Rwanda. School going children around the border attended both day and boarding schools. Day students cross the border daily because they have to return to their homes. Students from Rwanda cross to Uganda to attend schools in Kabale like Sainte Jerome Ndama

and Kisoro Vision Secondary School in Kisoro, and tertiary institutions like Kampala International University. Ugandan students cross to Rwanda to attend Kigali Green Hill Academy.

"...*The two schools I mentioned (Rise and Shine Primary School, and Kisoro Vision Secondary School) are where people from Rwanda send their children for education and they are boarding schools.....*" **KII Kisoro**

"...*people come from Kapchorwa District to Eldoret for school, other schools visited in Kenya by Ugandans in Lwakhakha border include; Lena Academy, Chepukui Primary and Secondary Schools and Namunde Primary School....*" **FGD P1 Lwakhakha**

**Social reasons.** People travel for social reasons including visiting family and friends and places of entertainment like bars, football fields on either side of the border. Some men have wives on both sides of the border (Kenya & Uganda) therefore they cross the border to visit relatives particularly in Bungoma and Mount Elgon.

"....*Others cross to drink alcohol in Uganda during the market days. Kenyans come to Uganda to drink because they feel beer in Uganda is cheap and waragi (local spirit) is illegal in Kenya.....*" **KII DSFP**

"..... *people move from Lwakhakha village in Uganda to Lwakhakha in Kenya for football games at the Lwakhakha government football pitch....*" **FGD P4 Lwakhakha**

**Geographical factors.** Participants reported several geographic factors influencing movement, these include mineral resources, topographical features, plants and animals in national parks and reserves.

"...*tourists move to see the mountain caves at Mgahinga National Park, the highest peak of the three mountains which is Muhabura Mountain and Bwindi impenetrable national park...*" **KII Cyanika**

"...*travelers move to Mubende from Rwanda for gold mining...*" **P7, FGD Cyanika**

**Movement patterns across the border.** Respondents reported using both official and unofficial PoEs for entry and exit for movement across the borders (Fig 1A and 1B). The official PoEs on the Uganda-Rwanda border include Cyanika, Katuna, and Mirama Hills; while PoEs on the Uganda-Kenya border include Lwakhakha, Suam, Busia, and Malaba. There are over 20 porous routes along the Namusindwa-Bungoma border. The most frequently used illegal routes between Kenya and Uganda are Soko Mujinga, Daraja ya Mungu, Mundidi, Chepkube, Bukhontso and Soono. Porous routes on the Uganda-Rwanda border included Mgahinga, Kibaya, Kanyamucucu, Rugabano, and Gatwe among others.

Unofficial PoEs were preferred because they had less or no restrictions: like absence of health screening, immigration check points which created a stop and were a suitable environment for smuggling.

Some of the respondents were quoted saying:

"...*People do not want to pass through the official border because they don't want to be tested for COVID-19 due to its high cost, and they do not want to be checked...*" **FGD P3 Cyanika**

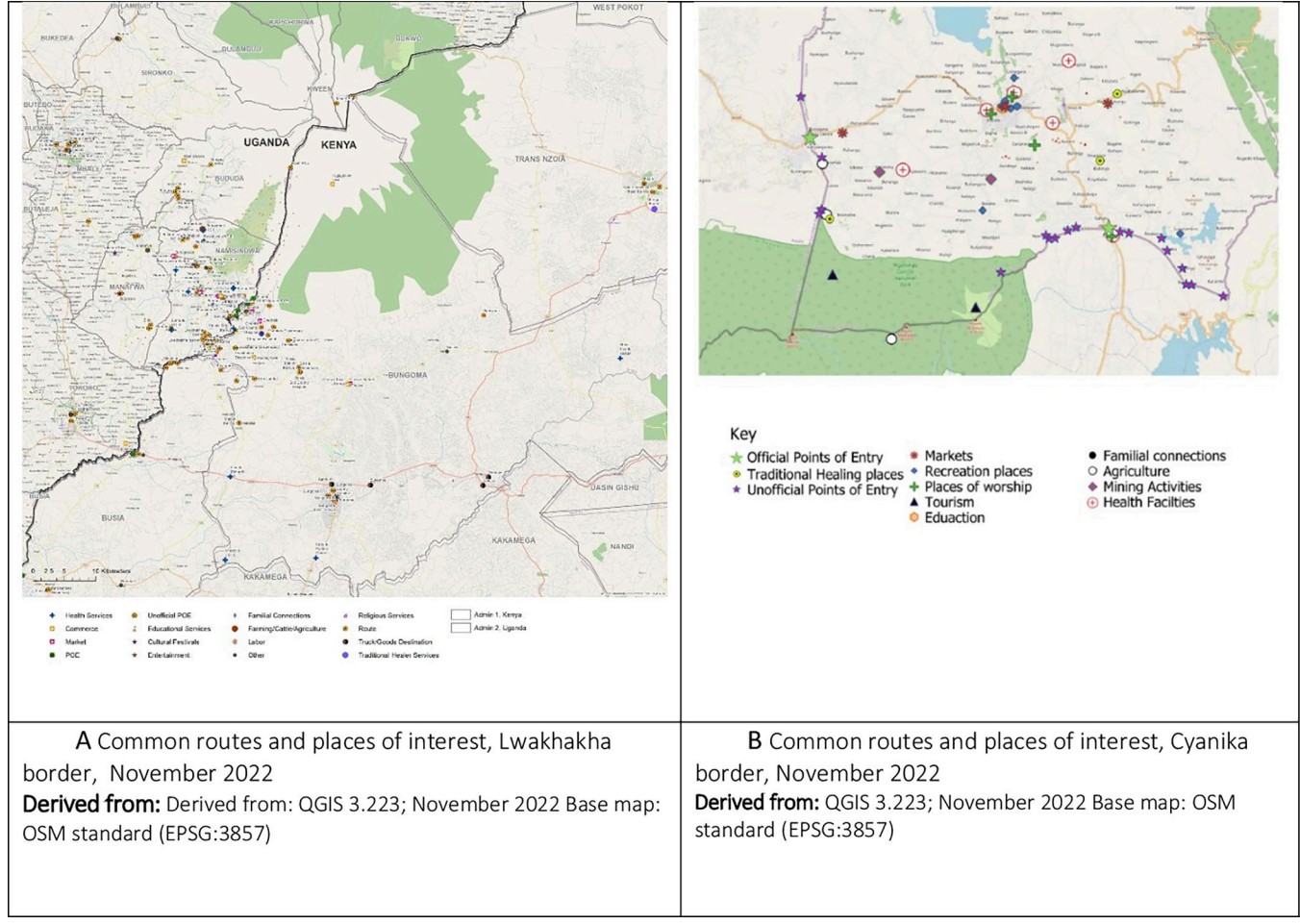

A Common routes and places of interest, Lwakhakha border, November 2022
**Derived from:** Derived from: QGIS 3.223; November 2022 Base map: OSM standard (EPSG:3857)

B Common routes and places of interest, Cyanika border, November 2022
**Derived from:** QGIS 3.223; November 2022 Base map: OSM standard (EPSG:3857)

**Fig 1. A** Common routes and places of interest, Lwakhakha border, November 2022. Derived from: Derived from: QGIS 3.223; November 2022 Base map: OSM standard (EPSG:3857). **B** Common routes and places of interest, Cyanika border, November 2022. Derived from: QGIS 3.223; November 2022 Base map: OSM standard (EPSG:3857).

*". . .the Lwakhakha border line is so porous; there are over 20 other routes through the border where people cross to either Uganda or Kenya because they do not have travel documents. . ..***"** **KII Lwakhakha**

*". . . . .Animals move from Nyagatare in Rwanda to Nyakabande animal market in Uganda while others go to Kyankwanzi and Nakasongola both in Uganda for grazing."* **FGD P5 Cyanika**

**Frequency of travel and duration of stay.** Respondents reported that the frequency of travel and duration of stay varied depending on the season and the reason for travel.

*". . .truck drivers cross the border daily; Women/mothers cross daily and weekly; Ugandans looking for employment cross weekly and monthly; Sudanese refugees cross weekly and monthly; Ugandans travelling for festivals of circumcision cross daily; Ugandan traders cross weekly every Tuesday and Saturday on market days. . . ."* **FGD P1 Lwakhakha**

*". . .truck drivers stay for about 6–12 hours as they load and wait for goods, 3–4 days for individual coming in to seek medical services and more depending on the illness being sought*

*treatment for, one day for traders who are just buying goods and going back to their homes or across the border and family visits which also depend on wish. . .female sex workers who go there on every Friday and come. . ."* **FGD P2 Cyanika**

**Volume of travellers and seasonality.** The volume of travellers varied across seasons and key events on the other side of the border such as market days. Differences were reported between the type point of entry used (official vs unofficial).

*". . .on average, 5000 persons pass through porous borders per market day, and around 2000 persons on non-market days for casual work. . ."* **FGD P1 Cyanika**

The peak times of travel were related to agricultural seasons, religious and cultural festivals.

*". . . From Uganda to Kenya movement is mainly January—April and June–August during cultivating and planting seasons. . . From Kenya to Uganda in December during cultural festivals. . .they generally move throughout the year but the months mentioned above have the most movements. . ."* **FGD P3 Lwakhakha**

*". . . travelers from Rwanda move to Uganda in June to look for jobs in Mubende district during the harvest season. . .In December they travel back to celebrate Christmas with their families. . .."* **KII Cyanika**

*". . ...Ugandans travel to Rwanda to Celebrate the new year's day. . . In Rwanda they celebrate better than in Uganda. . .. . ."* **KII Cyanika**

**Means of transport.** The participants that responded stated that travellers walk across, use motorcycles ("bodaboda"/ "tuktuk"), private and public vehicles and some are carried on the back to cross rivers.

*". . .People use bodaboda/ motorcycles, daily commuting buses/taxis and foot through unofficial borders to avoid health screening at the official PoE for fear of quarantine or isolation. . .at unofficial points people swim across the river or when the river is shallow, a guide holds the traveller's hand and they are guided to walk through the river. . ."* **FGD P4 Lwakhakha**

**Influence of on going EVD outbreak in Uganda on population movement.** Respondents reported that influence of the ongoing EVD outbreak in Uganda was different for each country. There was reduced movement to Uganda from Rwanda while movements cross the Uganda-Kenya border were not affected.*". . .Ebola outbreak has led the reduction of movements to Uganda because Rwanda has put more restrictions for travellers returning from Uganda. . . a traveller has to be tested every time he/she crosses border. . .security guards have been deployed along porous borders; and this movements reduction has caused total loss in business. . ."*

*". . .. people have continued to go about their business but with caution because we are at the border and anything is bound to happen*

Some cross border communities did not perceive EVD as a threat and therefore carried on with travel was to be very far from them; in Kampala and Mubende with no reason for worry. They reported that EVD had not scared them as much as coronavirus did.

*". . .The fact that Ebola has not yet reached Nyabitare village, it has not compromised people's movements across borders. . . some youths still travel to Mubende, an active outbreak area in search fro jobs in Agriculture. . ."* **P4 FGD Cyanika**

*". . .Some business men and truck drivers continue to travel to markets in Jinja and stay there for a couple of days during the ongoing EVD outbreak. . .."* **KII Lwakhakha**

**Points of interest and routes.** Points of interest included markets, places of worship, health facilities, education facilities and recreational/accommodation facilities (Fig 1A and 1B).

## Discussion

Human mobility across the border has the potential to accelerate the spread of infectious diseases across countries. We explored human mobility patterns along the Uganda-Kenya and Uganda-Rwanda borders during an ongoing Sudan EVD outbreak in Uganda in November 2022. Our findings indicated that communities travel across borders for livelihood, healthcare, religious, social, and cultural purposes. Key destination points of travellers included high-volume areas such as markets, health facilities, places of worship, entertainment/recreation venues, schools and busy towns in Uganda and Kenya with confirmed EVD cases. Travellers preferred to use unofficial Points of Entry where there's no health screening and registration services.

Our findings indicated a potential for disease transmission with travel for healthcare and risky sexual behavior. Ill travelers could potentially spread disease as observed in previous outbreaks such as the COVID-19 pandemic [6]. Risky behavior such as commercial sex and alcohol consumption could increase sexually transmitted diseases such as HIV and syphilis at the border [7, 8]. Behavioral change communication interventions targeted at risky behavior tailored to the needs of cross border communities.

We found that travellers sought key health services such as maternal health, child health (immunisation), and HIV services. Travel for HIV services across the border highlights the need of ensuring the HIV continuum of care around and across the border. Previous studies in the region have indicated the need for tailor made strategies to support linkages to HIV services across the border [9]. Similar to a previous study by Ssengooba et al ease of crossing the border, services being free, and availability of quality services facilitated seeking of health services across the border (10). Transnational coordination and service delivery of HIV treatment services could improve continuity of care [10].

Movement from Kenya and Rwanda to areas in Uganda where there was an ongoing EVD outbreak presented a potential risk of transmission of EVD to these countries. Communities reported frequent travel with relatively long periods of stay in Uganda presenting opportunities for more human-to-human interaction thus possible exposure to disease. Key destinations in Uganda included Mubende, Jinja, and Kampala, which had confirmed EVD cases [11]. In both destinations with confirmed EVD cases and those without, travellers moved to high volume sites including markets, places of worship, and entertainment which are usually characterised by low surveillance and poor implementation of prevention measures. Further, the reported preference to use unofficial Points of Entry and with no health screening and registration likely led to missed opportunities for case detection. Additionally, there are missed opportunities collecting information from travellers such as travel history and contact information which are key for case or contact tracing investigations. Previous studies have highlighted how the use of unofficial points of entry led to the spread of EVD in the West Africa EVD outbreak [12].

We highlighted a possible risk of transmission of zoonotic diseases due to animal trade and consequently movement of animals across the border along Uganda-Rwanda. Along the Uganda-Rwanda border, there is a risk of transmission of brucellosis, rift valley fever given that these diseases are endemic in Southwestern Uganda [13, 14].

### Study limitation

In this study participants were selected based on their availability and willingness to participate in the study therefore their views maybe different from those of the broader community thus limiting generalizability of the findings.

### Conclusion

In conclusion, complex population movement and connectivity patterns were identified along the border. Communities travelled to high-volume service areas and busy towns in Kenya, Rwanda, Democratic Republic of Congo, and Uganda for various reasons. Travellers preferred to use unofficial points of entry where there is no health screening and registration services. We recommend enhanced community-based surveillance in identified key high-risk areas such as markets, key destinations and porous borders. Behavioral change communication interventions targeted at risky behavior tailored to the needs of cross border communities and trans national coordination and service delivery of HIV treatment services could improve continuity of care in cross border communities.

### Public health actions

Following dissemination of our findings, border districts of the three countries resolved to revise their district emergency response and preparedness plans by strengthening community-based surveillance at key destinations points, unofficial and porous borders accounting for seasonality of travellers during the preparedness activities and strengthening capacities of those health facilities. Plans are underway to provide integrated HIV services across border areas with the main focus of ensuring HIV continuity of care.

### Supporting information

**S1 Checklist. Inclusivity in global research.**
(DOCX)

### Acknowledgments

We would like to thank the Ministry of Health of Uganda, Kenya, and Rwanda for the technical and financial support. We acknowledge the Uganda National Institute of Public Health, Kenya Field Epidemiology and Laboratory Training Programme, and Rwanda Field Epidemiology Training Programme for availing us with the technical staff that executed this project. We appreciate the technical support provided by the Border Health Unit in the Division of Surveillance, Information and Knowledge Management. We thank Kisoro and Namisindwa District Local Governments, and the port health teams at Cyanika and Lwakhakha for their participation in this activity. Finally, we thank the US-CDC for supporting the activities of the Uganda Public Health Fellowship Program (UPHFP); but specifically for supporting the Border health Program through the International Association of National Public Health Institutes (IANPHI).

## Author Contributions

**Conceptualization:** Patrick King, Mercy Wendy Wanyana.

**Data curation:** Joyce Owens Kobusinge.

**Formal analysis:** Patrick King, Mercy Wendy Wanyana, Brenda Nakafeero Simbwa, Marie Gorreti Zalwango.

**Investigation:** Patrick King, Harriet Mayinja, Brenda Nakafeero Simbwa, Joyce Owens Kobusinge, Benon Kwesiga, Serah Nchoko, Edna Salat, Freshia Waithaka, Oscar Gunya, Fredrick Odhiambo, Metuschelah Habimana, Gabriel Twagirimana, Ezechiel Ndabarinze, Alexis Manishimwe, Harriet Itiakorit.

**Methodology:** Patrick King, Mercy Wendy Wanyana.

**Project administration:** Alex Riolexus Ario.

**Resources:** Alex Riolexus Ario.

**Supervision:** Richard Migisha, Daniel Kadobera, Benon Kwesiga.

**Visualization:** Patrick King, Serah Nchoko, Edna Salat, Freshia Waithaka.

**Writing – original draft:** Patrick King, Marie Gorreti Zalwango.

**Writing – review & editing:** Patrick King, Mercy Wendy Wanyana, Harriet Mayinja, Brenda Nakafeero Simbwa, Joyce Owens Kobusinge, Richard Migisha, Daniel Kadobera, Benon Kwesiga, Lilian Bulage, Doreen Gonahasa, Peter Ahabwe Babigumira, Edna Salat, Freshia Waithaka, Oscar Gunya, Fredrick Odhiambo, Vincent Mutabazi, Metuschelah Habimana, Harriet Itiakorit, Samuel Kadivani, Katy Seib, Ellen Whitney, Alex Riolexus Ario.

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
