## [Decision Letter · Decision Letter 0]

30 Apr 2024

PGPH-D-24-00279

Cross Border Population Movement Patterns, Kenya, Uganda, and Rwanda, November 2022.

Dear Dr. Patrick King

Thank you for submitting your manuscript to PLOS Global Public Health. After careful consideration, we feel that it has merit but does not fully meet PLOS Global Public Health’s publication criteria as it currently stands. Therefore, we invite you to submit a revised version of the manuscript that addresses the points raised during the review process.

We look forward to receiving your revised manuscript.

Kind regards,

Wilber Sabiiti

Academic Editor

Journal Requirements:

1. In the ethics statement in the Methods, you have specified that verbal consent was obtained. Please provide additional details regarding how this consent was documented and witnessed, and state whether this was approved by the IRB"

2. Please include the following request in the decision letter, and ping me with follow-up. “Please include a complete copy of PLOS’ questionnaire on inclusivity in global research in your revised manuscript. Our policy for research in this area aims to improve transparency in the reporting of research performed outside of researchers’ own country or community. The policy applies to researchers who have travelled to a different country to conduct research, research with Indigenous populations or their lands, and research on cultural artefacts. The questionnaire can also be requested at the journal’s discretion for any other submissions, even if these conditions are not met.  Please find more information on the policy and a link to download a blank copy of the questionnaire here: https://journals.plos.org/globalpublichealth/s/best-practices-in-research-reporting. Please upload a completed version of your questionnaire as Supporting Information when you resubmit your manuscript.

3. Please ensure that Funding Information and Financial Disclosure Statement are matched.

4. In the Funding Information you indicated that no funding was received. Please revise the Funding Information field to reflect funding received.

5. Please provide separate figure files in .tif or .eps format only and remove any figures embedded in your manuscript file. Please also ensure all files are under our size limit of 10MB.

Additional Editor Comments (if provided):

Reviewers' comments:

Reviewer's Responses to Questions

**Comments to the Author**

1. Does this manuscript meet PLOS Global Public Health’s publication criteria? Is the manuscript technically sound, and do the data support the conclusions? The manuscript must describe methodologically and ethically rigorous research with conclusions that are appropriately drawn based on the data presented.

Reviewer #1: Yes

Reviewer #2: Yes

2. Has the statistical analysis been performed appropriately and rigorously?

Reviewer #1: N/A

Reviewer #2: N/A

3. Have the authors made all data underlying the findings in their manuscript fully available (please refer to the Data Availability Statement at the start of the manuscript PDF file)?

Reviewer #1: Yes

Reviewer #2: Yes

4. Is the manuscript presented in an intelligible fashion and written in standard English?

Reviewer #1: Yes

Reviewer #2: Yes

5. Review Comments to the Author

Reviewer #1: This is a good paper but I believe it is limited in several aspects

1. The is no association of public health factors with population movements

2. The times and periods when the movements are at peak or lowest are not clearly specified even though factors associated with the movements are indicated

3. It reads like the authors picked a section from a larger data set because this paper would be stronger if it had some quantitative data on surveillance that would include disease outbreaks and epidemics and how they impact population movement

4. I believe there should be a section on policy and governance and how it impacts population movements

5. The authors should provide a brief descriptive description of the study participants and how they were distributed, for example, how many were from Kenya, Uganda or Rwanda

6. Other than the mapping, the authors should specify the geographical factors that attract migrants/ mobile populations to specific sites.

7. I believe recommendations are not strong because critical data/information on public health is lacking

Reviewer #2: 1. Summary of the research and overall impression

Strengths Of the manuscript

This manuscript is based on an important study that tries to understand how population movement patterns are critical for informing cross-border disease control interventions. It provides population mobility patterns across the borders of the East African states: Kenya, Uganda, and Rwanda. The paper investigates how cross-border movement could influence the spread of infectious diseases, particularly during outbreaks like the Sudan EVD outbreak. Reasons for traveling across borders were assessed., including seasonal variations in the volume of travelers. Key destination points (points of interest) included: markets, health facilities, places of worship, education institutions, recreational facilities, and business towns.

The paper seems to conclude that there are complex patterns of population movement and connectivity across borders. The study found that travelers often used unofficial entry points, lacking essential health screening and registration services. This movement occurred despite the presence of formal border points. This finding underscores the importance of community-based surveillance at key destinations, unofficial entry points, and porous borders. The study has already informed policy and practice. Border districts in all three countries are revising their emergency response plans based on the study findings, and community-based surveillance at critical locations among others. The use of qualitative methods of data collection including participatory mapping using Population Connectivity Across Borders toolkits enhanced the richness of data.

Weakness of the Manuscript

The weakness of the manuscripts relies mainly on the connection of the population movement and the spread of the diseases in question. This should be explained further. It is not clear which questions were asked and what responses were provided in relation to for instance EVD.

Secondly, the use of grounded theory is important for this study but the authors seem not to explain in depth how this was utilized, issues of open coding, themes/categories, axial coding for making sense of data, and selective coding to help in the development of draft theory should be well elucidated.

2. Evidence and examples

Major Issues

Major issues that must be addressed in order for the manuscript to be improved and proceed to the next level:

a) The title needs refocusing: Cross Border Population Movement Patterns is quite broad. For clarity: It is not clear in the title what this cross-border population movement relates to. The authors need to revisit the title to reflect what is written in the main text of the manuscript. Is November 2022 important to be included in the title?

b) FGDs for business persons- it is not clear whether these were separated by gender (Male, female ). Please explain what was done since the authors indicated that for the other categories of FGDs the participants were male but for the business person they included females as well.. If the GFDs were disaggregated by gender it would be important to know what were the key issues emanating from the female FGDs compared to the male FGDs,

c) Ethics: It is indicated that the study was reviewed by the United States Centers for Disease Control and Prevention (U.S CDC) human subjects review board and consistent with applicable federal law and CDC policy. What about REC from Uganda/Kenya/Rwanda? And for Uganda was the study submitted to UNCST for approval from the UNCST? Or this was unnecessary.

Movement from Kenya and Rwanda to areas in Uganda where there was an ongoing

d) In the discussion section there is a lot of writeup on EVD outbreaks presenting a potential risk of transmission of EVD to these countries. However, it is not well presented in the findings. Please revisit the transcripts (FGDS and KII) to get some quotes to give us the voices of the participants related to their concerns about such diseases. But their lack of concern is also a finding.

Minor issues

Several typos are evident and these need to be addressed. Examples include the abbreviation of focus group discussions is FGD. But in some cases, the authors abbreviate it as FDG.

Overall Assessment

This manuscript's findings go beyond the value of a single East African state. It offers insights that can improve cross-border population mobility, surveillance, and disease control across the region. The manuscript can be published after effecting/addressing the changes indicated above.

6. PLOS authors have the option to publish the peer review history of their article (what does this mean?). If published, this will include your full peer review and any attached files.

**Do you want your identity to be public for this peer review?** For information about this choice, including consent withdrawal, please see our Privacy Policy.

Reviewer #1: No

Reviewer #2: **Yes: **Stella Neema

---

## [Editor Report · Decision Letter 1]

15 Aug 2024

Cross border population movements across three East African states: Implications for disease surveillance and response

PGPH-D-24-00279R1

Dear Mr Patrick King

We are pleased to inform you that your manuscript 'Cross border population movements across three East African states: Implications for disease surveillance and response' has been provisionally accepted for publication in PLOS Global Public Health.

Best regards,

Wilber Sabiiti

Academic Editor